# FIDELITY OF INTERPRETABILITY METHODS AND PERTURBATION ARTIFACTS IN NEURAL NETWORKS

**Lennart Brocki & Neo Christopher Chung**
Institute of Informatics
University of Warsaw
Banacha 2, 02-097 Warsaw, Poland
`{brocki.lennart,nchchung}@gmail.com`

## ABSTRACT

Despite excellent performance of deep neural networks (DNNs) in image classification, detection, and prediction, characterizing how DNNs make a given decision remains an open problem, resulting in a number of interpretability methods. Post-hoc interpretability methods primarily aim to quantify the importance of input features with respect to the class probabilities. However, due to the lack of ground truth and the existence of interpretability methods with diverse operating characteristics, evaluating these methods is a crucial challenge. A popular approach to evaluate interpretability methods is to perturb input features deemed important for a given prediction and observe the decrease in accuracy. However, perturbation itself may introduce artifacts. We propose a method for estimating the impact of such artifacts on the *fidelity* estimation by utilizing model accuracy curves from perturbing input features according to the Most Import First (MIF) and Least Import First (LIF) orders. Using the ResNet-50 trained on the ImageNet, we demonstrate the proposed fidelity estimation of four popular post-hoc interpretability methods.

## 1 INTRODUCTION

Deep learning has recently demonstrated state-of-the-art performance in computer vision tasks. However, their decision (detection, classification, and prediction) is difficult to interpret and explain. One of the major approaches to interpretability is to quantify the importance of input features with respect to the model's prediction. However, rigorous evaluation of those feature importance estimators is problematic due to the complexity of DNNs and the lack of ground truth. Here, we investigate challenges arising from a perturbation-based evaluation and estimate the *fidelity* of importance estimators using the Most Import First (MIF) and the Least Import First (LIF) perturbation curves.

In DNNs, back-propagation and its variations are often used to quantify importance scores of input features, which are also called saliency maps and pixel attribution (Baehrens et al., 2010; Simonyan et al., 2014). Developments of importance estimators are often justified with human-centered arguments (Kim et al., 2019; Springenberg et al., 2015), such as regions of interest (ROI) (Saporta et al., 2022; Brocki et al., 2022) and sparsity (Chalasani et al., 2020). An objective evaluation of interpretability methods is therefore very desirable. One of the most established approaches is to mask input features with high importance scores and to measure the degradation of prediction accuracy of the model (Samek et al., 2016; Zeiler & Fergus, 2014). However, in such perturbation-based evaluation approaches, it has been unclear whether the observed accuracy degradation stems from information removal or perturbation artifacts. Unnatural perturbations introduce artifacts such that the test set consisted of perturbed images deviates substantially from the training set and is therefore out-of-distribution (Dabkowski & Gal, 2017; Nalisnick et al., 2018; Qiu et al., 2021).

In this study we are focusing on disentangling the sources of decrease of model accuracy. To quantify the influence that artifacts have on the decrease of model accuracy, we conduct a series of computational experiments, based on measuring the change in model accuracy after feature perturbation in the Most Import First (MIF) or the Least Import First (LIF) order.

## 2  METHODS

For a given importance estimator, accuracy curves are calculated by first perturbing images $\mathbf{X}^i$ by blurring them (appendix A.1.1) according to corresponding masks $\mathbf{M}^i(\mathbf{X}^i)(n)$, which are obtained by ranking the input pixels according to their estimated importance scores and selecting a fraction $n$ of either MIF or LIF pixels (appendix A.1.2). The resulting perturbed images are fed to the model and then the model accuracy is measured(Samek et al., 2016). We quantify the MIF and LIF accuracy curves by the areas above the respective curves up to a fraction $n$ of perturbed pixels, which are denoted by $F(n)$ and $U(n)$ (Illustrated in fig. S4, and algorithm 1).

We assume that $F(n)$ consists of two components namely the accuracy decrease due to information removal $F_I(n)$ and due to artifacts $F_A(n)$, the same is assumed for $U(n)$. We propose to estimate $F_A(n)$ by

$$F_A(n) \leq \delta(n) = U(n) + F^s(n) - U^s(n) \tag{1}$$

where the superscript $s$ indicates that the images $\mathbf{X}$ have not been perturbed according to the corresponding masks $\mathbf{M}(\mathbf{X})$ but instead using $\mathbf{M}(\mathbf{X}')$, where $\mathbf{X}'$ are randomly chosen images. We thus compute two sets of accuracy curves, one using corresponding masks $\mathbf{M}(\mathbf{X})$ to obtain $U$ and the other using $\mathbf{M}(\mathbf{X}')$ obtain $F^s - U^s$.

The idea behind using random images is that in the expression $\Delta = F^s - U^s = F_I^s + F_A^s - U_I^s - U_A^s$ we can assume $F_I^s \approx U_I^s$ since the masks $\mathbf{M}(\mathbf{X}')$ do not contain meaningful information about the importance of pixels in $\mathbf{M}(\mathbf{X})$. If we furthermore make the natural assumption that, when averaged over all considered images, the strength of artifacts is not affected by perturbing $\mathbf{X}$ with $\mathbf{M}(\mathbf{X}')$ instead of $\mathbf{M}(\mathbf{X})$ it follows that $F_A^s \approx F_A, U_A^s \approx U_A$ and we obtain $\Delta \approx F_A - U_A$. Solving this for $U_A$ and substituting in $U_A \leq U$ one arrives at eq. (1), which implies together with $F_A \leq F$ that $F_I$, the relevant quantity to compare importance estimators, is in the interval

$$F - \delta \leq F_I \leq F. \tag{2}$$

For more details on this derivation see appendix A.1.3. We applied the proposed methods to four importance estimators, namely vanilla gradient (VG) (Baehrens et al., 2010; Simonyan et al., 2014), Integrated Gradient (IG) (Sundararajan et al., 2017), Smoothgrad (SG) (Smilkov et al., 2017), and Squared SmoothGrad (SQ-SG) (Hooker et al., 2019). See details in A.1.4.

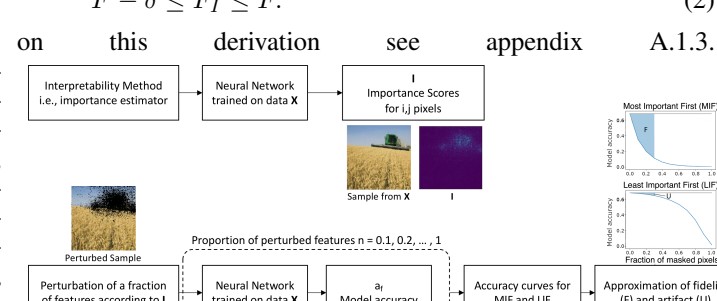

Figure 1: Diagram of the proposed *fidelity* evaluation method.

## 3  RESULTS AND CONCLUSIONS

The final results for the fidelity of importance estimators are given in table 1 for $n = 0.2$ and in table S1 for $n = 0.4$. The central result is that the measurement errors $\delta$ due to perturbation artifacts are small enough so that a meaningful ranking of importance estimators can be established. At $n = 0.4$ the bulk of the model accuracy is already gone except for the random estimator (table S1); nonetheless, the results agree with fidelity estimation with $n = 0.2$.

Comparing the estimated intervals for $F_I$ using equation 2 and the results from table 1, it can be seen that in both cases all estimators outperform a random ranking. The SG estimator performs the best. In the case of $n = 0.4$, performance of SG is on par with that of SQ-SG within the measurement uncertainty. This ranking aligns with human intuition since SG and SQ-SG appear to focus most closely on the object of interest (figure S1).

Table 1: Fidelity of importance estimators measured by $F$ for $n = 0.2$

| ESTIMATOR | $F \times 10^2$ | $\delta \times 10^2$ |
|---|---|---|
| VG | 3.8 | 0.4 |
| IG | 4.8 | 0.3 |
| SG | 6.0 | 0.2 |
| SQ-SG | 5.7 | 0.2 |
| RANDOM | 2.1 | 0.8 |

ACKNOWLEDGEMENTS

This work was funded by the ERA-Net CHIST-ERA grant [CHIST-ERA-19-XAI-007] long term challenges in ICT project INFORM (ID: 93603), by National Science Centre (NCN) of Poland [2020/02/Y/ST6/00071]. This research was carried out with the support of the Interdisciplinary Centre for Mathematical and Computational Modelling University of Warsaw (ICM UW) under computational allocation no GDM-3540 and the NVIDIA Corporation's GPU grant.

URM STATEMENT

The first author meets the URM criteria of ICLR 2023 Tiny Papers Track.

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

## A  APPENDIX

### A.1  SUPPLEMENTARY METHODS

#### A.1.1  PERTURBATION METHOD

We investigate how different perturbation methods affect model accuracy. Perturbing parts of images may introduce artifacts and create samples in evaluation that are not in the original distribution (Dabkowski & Gal, 2017; Nalisnick et al., 2018; Qiu et al., 2021). However, the impact of perturbation was rarely measured directly. In order to empirically disentangle the removal of information and introduction of artifacts, we run an experiment to quantify perturbation artifacts.

The pre-trained Tensorflow-Keras implementation of ResNet-50 (He et al., 2016), with an input dimension of $224 \times 224$ pixels is used with the ImageNet validation dataset (Deng et al., 2009). For each sample $\mathbf{X}^i$, we apply the following perturbation method:

- Blur($\sigma$): replaces pixel values with the corresponding values obtained by blurring the image with a Gaussian filter with radius $\sigma$.

The perturbed image $\mathbf{P}^i$ is obtained from the original image $\mathbf{X}^i$ as follows

$$\mathbf{P}^i = \mathbf{X}^i \odot \mathbf{M}^i + \mathbf{A}^i \odot (\mathbf{1} - \mathbf{M}^i), \tag{3}$$

where $\odot$ is the element-wise multiplication and $\mathbf{A}^i$ is an alternative image obtained through one of the perturbation methods. $\mathbf{M}^i$ is a mask with the same dimensions as $\mathbf{X}^i$ with value 0 for pixels to be replaced and 1 for pixels to be left invariant. When clear in context, the superscript $i$ for the sample is omitted.

When considering an importance estimator, the mask $\mathbf{M}^i(n)$ is obtained by ranking the features (e.g., input pixels) according to their estimated importance scores $\mathbf{I}_{i,j}$ and selecting a fraction $n$ of features in either the Most Import First (MIF) or Least Import First (LIF) order. These masks are

used to construct accuracy curves (see the next subsection) with an increasing amount of perturbed features.

When choosing a perturbation method, one faces a trade-off between effectively removing information and introducing artifacts. This can be clearly demonstrated for the blurring method. A slight blurring of an image (with a small $\sigma$) will introduce almost no artifacts but at the same time barely removes any information. Increasing the radius $\sigma$ of the Gaussian filter will remove more information but is also more likely to create artifacts. With a very large $\sigma$, a blurred image resembles a constant colored image. In order to select $\sigma$, we applied blurring with different radii of a Gaussian filter on the ImageNet samples and measured the model accuracy fig. S3 and with the minimum requirement that the accuracy should drop to zero when the images are completely blurred we choose $\sigma = 14$.

### A.1.2 QUANTIFICATION OF MIF AND LIF ACCURACY CURVES

Accuracy curves after feature perturbation for an image $\mathbf{X^i}$ are obtained by first masking an increasing fraction $n$ of input pixels according to $\mathbf{M^i}(n)$ in either MIF or LIF order, and then measuring the resulting model accuracy (Samek et al., 2016). This is equivalent to plotting $\mathbf{a}$ with $N = 1.0$ in algorithm 1 with $\mathbf{I}^i$ sorted in either a descending order (MIF) or an ascending order (LIF). See examples in figs. 1 and S4. The mask $\mathbf{M^i}(n)$ depends on the importance estimator under evaluation. For a given importance estimator, we create MIF and LIF curves for all samples in the validation set which was held out during training ($\mathbf{X^i}$ for i $= 1, \ldots, V$). We quantify the MIF and LIF accuracy curves by the areas above the respective curves up to a fraction $n$ of perturbed pixels, which are denoted by $F^{p,e}(n)$ and $U^{p,e}(n)$ (illustrated in fig. S4), where $p$ denotes the perturbation method and $e$ the importance estimator.

---

**Algorithm 1** Fidelity Estimation from MIF and LIF accuracy curves

---

**Input:** DNN $f(\mathbf{X})$; $n$ maximum fraction of perturbed pixels;
 Samples $\mathbf{X}^i_{W \times H \times C}$; Importance scores $\mathbf{I}^i_{W \times H}$ ($i = 1, \ldots, V$)
 **for** $n' \leftarrow 0$ to $n$ with a set increment of the sequence
  **for** $i \leftarrow 0$ to $V$
   $\mathbf{M}^i(n') = \mathbf{1}_{W \times H \times C}$
   $\mathbf{k}^i \leftarrow \text{sort(flatten}(\mathbf{I}^i))$
   (i.e. sort in descending order for MIF or ascending order for LIF)
   **for** $j \leftarrow 0$ to $WH$
    **if** $j < n'WH$ **then**
     $x, y \leftarrow$ pixel position of $\mathbf{k}^i_j$
     $\mathbf{M}^i(n')_{x,y} \leftarrow 0$
   $\mathbf{P}^i = \mathbf{X}^i \odot \mathbf{M}^i(n') + \mathbf{A}^i \odot (\mathbf{1} - \mathbf{M}^i(n'))$
   Apply DNN $f(\mathbf{P}^i)$, save classifications in $\mathbf{o}_{(n')}$
  Calculate the model accuracy using $\mathbf{o}_{(n')}$, $a_{(n')} = \frac{\text{\# Correct Predictions}}{\text{\# Total Predictions}}$
 $F$ = Area above the MIF accuracy curve traced by $\mathbf{a}$
 $U$ = Area above the LIF accuracy curve traced by $\mathbf{a}$
  (e.g., visualized in fig. S4)

---

Note that $\mathbf{O}$ is a matrix of DNN classifications, where $\mathbf{o}_{(n')}$ is a vector of length $V$ consisted of predicted classes for samples with $n'$ perturbation level. $\mathbf{a}$ is a vector of model accuracies at varying levels perturbation ($n'$), which are plotted as MIF/LIF accuracy curves.

We decompose $F$ and $U$ as follows

$$F^{p,e}(n) = F_I^{p,e}(n) + F_A^{p,e}(n)$$
$$U^{p,e}(n) = U_I^{p,e}(n) + U_A^{p,e}(n),$$

where the subscript $I$ refers to the information removal component ($F_I$ and $U_I$) and the subscript $A$ refers to the perturbation artifact component ($F_A$ and $U_A$). When clear in context, $n$, $e$, or $p$ may be omitted for brevity. The components $F_A$ and $U_A$, by definition, capture any decrease in model accuracy that is not due to the removal of information relevant to the model. In general, only $F$ and

$U$ can be directly measured, and the decomposition into the information and artifact contribution is in general unknown. Details on obtaining $F$ and $U$ are provided in algorithm 1. This decomposition is motivated by the fact that even the masking of non-informative pixels can lead to a drastic decrease in accuracy (Fong & Vedaldi, 2017), which is also leveraged in adversarial attacks (Kurakin et al., 2018).

We assume that an information removal is non-negative, $F_I \geq 0$ and $U_I \geq 0$, such that

$$F_A \leq F$$
$$U_A \leq U. \tag{4}$$

### A.1.3 ESTIMATION OF PERTURBATION ARTIFACTS

The decomposition of $F$ into $F_A$ and $F_I$ is unknown and our goal is to estimate it. Consider the following expression for a certain proportion of perturbed features $n$ and a given perturbation method $p$,

$$\Delta^e = F^e - U^e$$
$$= F_A^e + F_I^e - U_A^e - U_I^e, \tag{5}$$

where $e$ can be any importance estimator.

To demonstrate the basic idea of our method of obtaining an estimate for $F_A$, let us consider the case where $\mathbf{M}(\mathbf{X})$ is obtained from randomly assigned importance scores. Since the order in which pixels are masked is completely random we can assume that $F_I^r = U_I^r$ from which it follows that

$$\Delta^r = F_A^r - U_A^r, \tag{6}$$

where the superscript $r$ indicates random ranking of pixels. Solving equation 6 for $U_A^r$ and using equation 4, we find

$$F_A^r \leq \Delta^r + U^r. \tag{7}$$

Since the quantities on the right hand side can readily be determined, this provides for us an upper bound for $F_A^r$.

While the artifact contribution for a random baseline can be easier to analyze, we are interested in $F_A^e$, where $F_I^e \neq U_I^e$. Since the cancellation of $F_I^e$ and $U_I^e$ in the expression for $\Delta^e$ is crucial, the proposition 6 can not be applied to a meaningful importance estimator. To remedy this situation, we perform an experiment in which we are perturbing images $\mathbf{X}$ with a non-informative mask $\mathbf{M}^s(\mathbf{X}')$. $\mathbf{M}^s(\mathbf{X}')$ is a mask that belongs to a randomly chosen $\mathbf{X}' \neq \mathbf{X}$ and is further shifted horizontally and vertically by random amounts between 10 and 100 pixels. Shifting removes a bias in the ImageNet dataset in which labeled objects tends to be in the center. Thus, we can expect that the importance ranking in $\mathbf{M}^s(\mathbf{X}')$ is random and $F_I^{s,e} \approx U_I^{s,e}$, where the superscript $s$ in $F_I$ and $U_I$ indicates that $\mathbf{M}^s(\mathbf{X}')$ have been used.

We now define a new delta

$$\tilde{\Delta}^e = F^{s,e} - U^{s,SQ-SG}, \tag{8}$$

where $SQ - SG$ refers to squared SmoothGrad. We choose $U^{s,SQ-SG}$ because $U^{SQ-SG}$ is the smallest value among the considered importance estimators (figure S2, right side), which allows us to constrain our estimate of the strength of perturbation artifacts ($F_A^e$) more strongly, see eq. (10) below. Using $F_I^{s,e} \approx U_I^{s,SQ-SG}$, we can write equation 8 approximately as

$$\tilde{\Delta}^e \approx F_A^{s,e} - U_A^{s,SQ-SG}$$
$$\approx F_A^e - U_A^{SQ-SG}, \tag{9}$$

where in the second line we have assumed that $F_A^{s,e} \approx F_A^e$ and $U_A^{s,SQ-SG} \approx U_A^{SQ-SG}$. The rationale behind this assumption is that the introduced artifacts are not specific to the image as compared to, for example, the adversarial artifacts found in (Szegedy et al., 2013) which are obtained through optimization. We, therefore, expect that any artifact-inducing patterns present in the masks $\mathbf{M}(\mathbf{X})$ are, on average, preserved when using $\mathbf{M}^s(\mathbf{X}')$ instead.

Solving equation 9 for $U_A^{SQ-SG}$ and substituting in equation 4, with $e = SQ - SG$, we finally obtain

$$F_A^e \leq \max(\tilde{\Delta}^e, 0) + U^{SQ-SG}, \tag{10}$$

where we additionally ignore negative $\tilde{\Delta}^e$ to obtain a more conservative estimate. The minimum estimate for the bound on $F_A^e$ is, therefore, $U^{SQ-SG}$, which would be the exact upper bound for $F_A^e$ if the strength of artifacts depended solely on $n$. This is because in that case, $F_A^e(n) = U_A^{SQ-SG}(n)$ and from equation 4, one finds $F_A^e \leq U^{SQ-SG}$. With the additional term $\tilde{\Delta}^e$ equation 10 we estimate variations between $F_A^e(n)$ and $U_A^{SQ-SG}(n)$ that arise due to the particular distribution of perturbed pixels when masking MIF or LIF and for different estimators.

The estimated upper limit equation 10 is understood as a systematic measurement error

$$\delta^e(n) = U^{SQ-SG}(n) + \max(\tilde{\Delta}^e(n), 0) \tag{11}$$

that can reduce the measured fidelity due to artifacts. Thus, we estimate $F_I^e$, which is the relevant quantity to compare importance estimators, to be in the interval

$$F^e - \delta^e \leq F_I^e \leq F^e. \tag{12}$$

### A.1.4 IMPORTANCE ESTIMATORS

We apply the proposed method of calculating fidelity and estimating artifacts to four importance estimators. The estimators under comparison are:

- **Vanilla gradient (VG)** (Baehrens et al., 2010; Simonyan et al., 2014): Gradients of the class score $S_c$[1] with respect to input pixels $x$

$$\mathbf{e} = \frac{\partial S_c}{\partial x}$$

- **Integrated gradient (IG)** (Sundararajan et al., 2017): Average over gradients obtained from inputs interpolated between a reference input $x'$ and $x$

$$\mathbf{e} = \left(x - x^0\right) \times \sum_{k=1}^{m} \frac{\partial S_c\left(x^0 + \frac{k}{m}\left(x - x^0\right)\right)}{\partial x} \times \frac{1}{m},$$

  where $x'$ is chosen to be a black image and $m = 25$.

- **Smoothgrad (SG)** (Smilkov et al., 2017): Average over gradients obtained from inputs with injected noise

$$\mathbf{e} = \frac{1}{n} \sum_{1}^{n} \hat{\mathbf{e}}\left(x + \mathcal{N}\left(0, \sigma^2\right)\right),$$

  where $\mathcal{N}\left(0, \sigma^2\right)$ represents Gaussian noise with standard deviation $\sigma$, $\hat{\mathbf{e}}$ is obtained using vanilla gradient and $n = 15$.

- **Squared SmoothGrad (SQ-SG)** (Hooker et al., 2019): Variant of SmoothGrad that squares $\hat{\mathbf{e}}$ before averaging

$$\mathbf{e} = \frac{1}{n} \sum_{1}^{n} \hat{\mathbf{e}}\left(x + \mathcal{N}\left(0, \sigma^2\right)\right)^2.$$

IG has been introduced to overcome the issue of VG possibly ignoring important features if the gradients flatten at the input. This effect could lead VG to focus on irrelevant features and might be a cause for noisy saliency maps. SG also tackles the problem of noisy saliency maps by averaging VG over a set of inputs with added noise which, interestingly, leads to more focused saliency maps.

All described methods create three-dimensional saliency maps and to obtain two-dimensional ones, we sum up the absolute value of the color channels.

---

[1]The class score $S_c$ is the activation of the neuron in the prediction vector that corresponds to the class $c$

## B SUPPLEMENTARY FIGURES AND TABLES

The systematic error and the fidelity are functions of $n$, and one has to decide which value to choose. We are going to evaluate the fidelity using $n = 0.2$ and $n = 0.4$, using four importance estimators. At $n = 0.4$ the bulk of the model accuracy is already gone except for the random estimator. From figure S2 one can see that beyond $n = 0.4$, $U$ starts to grow considerably which would lead to very large measurement errors. The final results for the fidelity of importance estimators are given in table 1 for $n = 0.2$ and in table S1 for $n = 0.4$. The central result is that the measurement errors due to perturbation artifacts are small enough so that a meaningful ranking of importance estimators can be established.

Comparing the estimated intervals for $F_I$ using equation 12 and the results from table 1, it can be seen that in both cases all estimators outperform a random ranking. The SG estimator performs the best. In the case of $n = 0.4$, performance of SG is on par with that of SQ-SG within the measurement uncertainty. This ranking aligns with human intuition since SG and SQ-SG appear to focus most closely on the object of interest (figure S1).

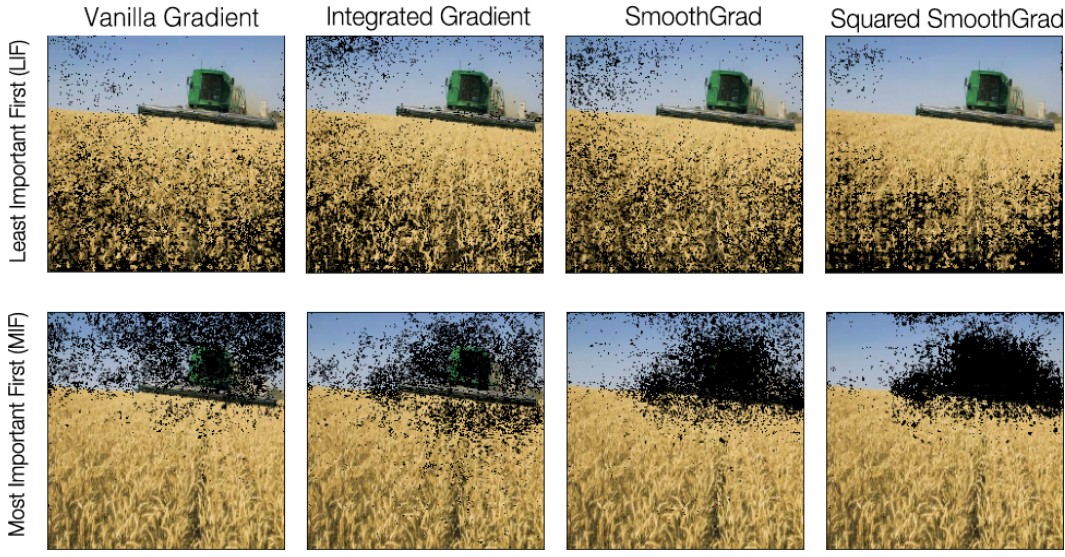

Figure S1: Examples for masks $\mathbf{M}$ obtained using different importance estimators. For $n = 0.2$, $20\%$ of pixels are perturbed in MIF or LIF orders. See the section A.1.4 for definitions of the importance estimators.

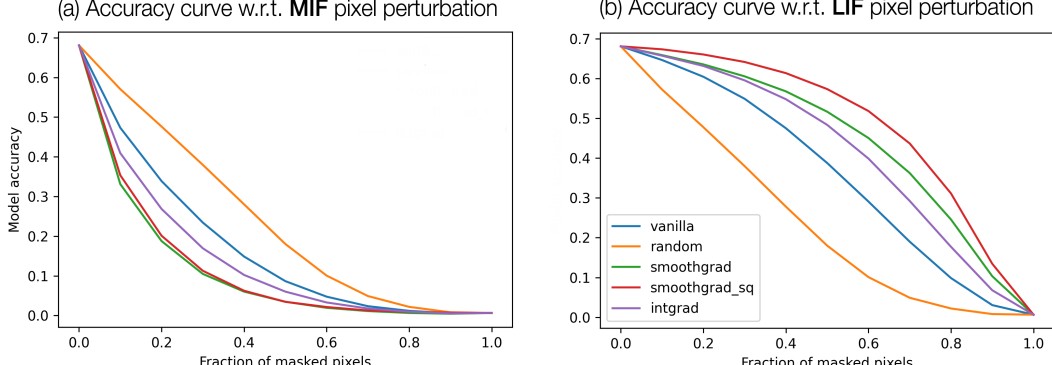

Figure S2: Comparison of decrease in model accuracy when pixels are masked in MIF (*left*) or LIF (*right*) order, according to different importance estimators (same color labels). The ResNet-50 trained on the ImageNet validation dataset are used to obtain the prediction and accuracy (Section A.1.1).

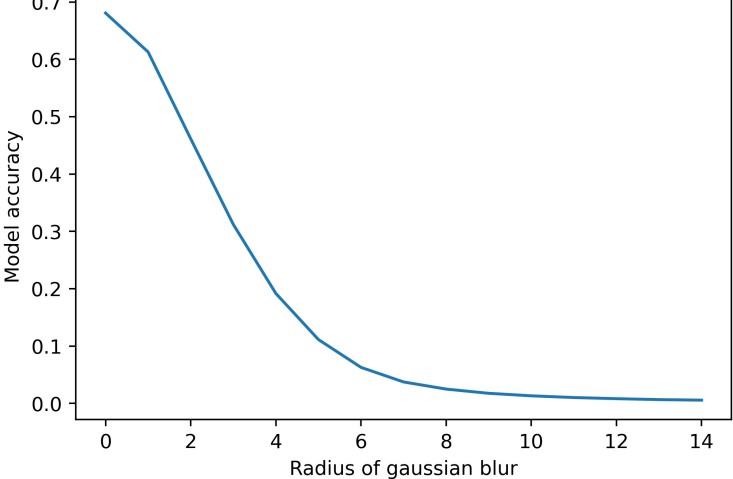

Figure S3: Average model accuracy of ResNet-50 on the ImageNet, with an increasing strength (radii $\sigma$) of blurring used for the whole input images.

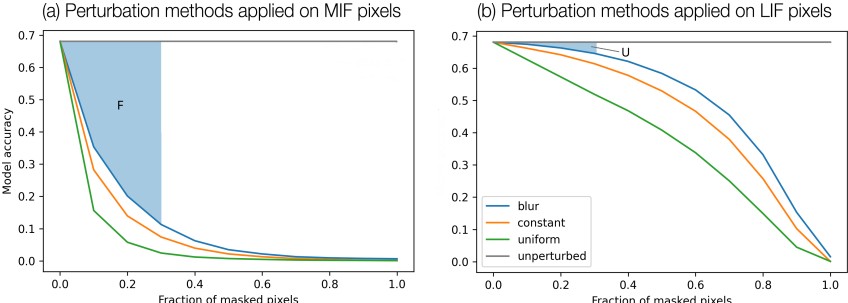

Figure S4: Comparison of decrease in model accuracy over the fraction of masked pixels when MIF (*left*) or LIF (*right*) pixels are perturbed using different methods. The ranking of the importance of the pixels is according to the squared SmoothGrad estimator. Area $F(n)$ and $U(n)$ quantify the accumulated decrease in model accuracy when some fraction $n$ of MIF and LIF pixels is perturbed, respectively. In blue, $F(0.3)$ and $U(0.3)$ are shown.

Table S1: Fidelity of importance estimators measured by $F$ for $n = 0.4$ with a systematic error $\delta$ determined by the right-hand side of equation 10 and using blur as perturbation method.

| ESTIMATOR | $F \times 10^2$ | $\delta \times 10^2$ |
|---|---|---|
| VG | 12.6 | 1.7 |
| IG | 14.8 | 1.6 |
| SG | 17.3 | 1.0 |
| SQ-SG | 16.8 | 1.0 |
| RANDOM | 8.1 | 3.0 |

