# OpenReview forum: "Fidelity of Interpretability Methods and Perturbation Artifacts in Neural Networks"
_ICLR.cc/2023/TinyPapers — Submitted to Tiny Papers @ ICLR 2023_

### Official Review · Reviewer_GRQN · 2023-03-31

**Confidence:** 4

**Summary Of Contributions:**

The authors present a fidelity measure to evaluate the correctness of the explanations provided by the different saliency methods. They propose to study trends of decease in black box model accuracy after removing the features in most-important-first (MIF) and least-important-first (LIF) manner.

**Rating:**

High Potential (HP): a submission which meets the reviewing criteria and has potential to make an impact on the field

**Strengths And Weaknesses:**

Strengths:

1. The proposed method is simple and straightforward to use.
2. It can be used without retraining of the black box model.
3. An important ambiguity in understanding saliency based XAI methods is that, if we remove important features generated by the XAI method and we observe a decrese in accuracy, then we are not sure if the drop in accuracy happens due to removal of important features or becuase the samples become unnaturally distorted (Out-of-distribution(OOD)). The paper presents an interesting approach to separately measure the drop in accuracy after removing important and unimportant features. The drop in accuracy after removing the unimportant features gives an indication of drop due to OOD.

Weaknesses:
1. The results could have been better reported. For example, the plot S2 and S4 are showing the accuracy decays for all the images in the validation set, as written in caption. However, the plots seem to be drawn for a single image, and the standard deviation curve should have been also reported around the mean curves. Generally, we can not infer much from a plot generated using single image or without seeing the standard deviations if the plot is generated from multiple images.

**Suggested Changes:**

1. The hypothesis on which the work stands, assumes that most-important-first (MIF) and least-important-first (LIF) based feature removal startegies results in similar OOD data. However, for the images where there is one dominant foreground object and rest of the image is background, (figure S1), the most-important-first (MIF) results into a big sized concentrated removal region (that greatly degrades the naturalness of the image) and least-important-first (LIF) removal results in sparse small distortions (which doesn't affect the naturalness of the image much). Hence, to further validate the approach, images with multiple dominant foreground objects (an image with a cat and a dog) can be experiment with. In that case, remove the important regions on dog and check the drop in prediction probabilty of cat. This will give an idea if least-important-first (LIF) agrees with it.
2. You should provide results for perturbation based XAI approaches, for example, LIME(Ribeiro et al 2016), RISE(Petsiuk et al 2018) etc. Currently, the results include only gradient based approaches.

---

### Official Review · Reviewer_xR4m · 2023-04-01

**Confidence:** 3

**Summary Of Contributions:**

The authors propose a way to evaluate the reliability of methods for identifying the most important features (here, pixels) in a sample (here, images).

**Rating:**

Great Start (GS): a submission which meets some of the reviewing criteria but has room for improvement

**Strengths And Weaknesses:**

S: The problem is relevant, and the method seems interesting.

W: The paper is confusing and should be better proofread. While  the general idea behind the method can be  guessed, the exact method is hard  to figure out given the lack of information.

**Suggested Changes:**

- Section 2 introduces U and F quantities.

- But in section 3, U  disappears. Instead we have F and an unexplained "delta" in  Table 1.

- What is delta?

- What happened to U? If F is sufficient, why did  we need to compute U?

- Looking at section 2 and trying to fill in the blanks,  the idea seems to be that, for a given n,  the difference between F and U should represent how valid is the "importance" of the pixels calculated by the method, since they both involve perturbing the same amount of pixels, and the only difference is the importance order (but  see the objection raised by reviewer GRQN). But this is never explicitly stated. We are not given an intuitive explanation of why we should be interested in F and U, or why U disappears after Section 2. Please provide at least a high-level intuitive explanation.

- There seems to be a complex, lengthy justification in the Appendix. That's not sufficient. Explain what your method actually  is in the main text!

---

### Official Review · Reviewer_74sd · 2023-04-02

**Confidence:** 5

**Summary Of Contributions:**

The paper claims to propose a new evaluation metric to assess the fidelity of existing attribution-based post-hoc interpretability methods. They have conducted experiments using ResNet-50 trained on ImageNet and four existing interpretability methods.

**Rating:**

Needs Clarification (NC): a submission which does not meet the reviewing criteria and needs clarification for its described problem or solution

**Strengths And Weaknesses:**

Strengths of the paper

* Assessing fidelity of the explanations produced by interpretability methods is a crucial step for the developments of interpretability domain. Further, it is true that the widely used perturbation-based approach may cause the images to deviate from the training data and result in drop of accuracy merely due to this deviation. Hence, the paper tries to solve an important issue in interpretability domain.

* The authors have mentioned that they conducted experiments using multiple perturbation methods and multiple interpretability methods.
The paper follows the basic requirements of the conference. However, it seems like the anonymity of the paper is violated a bit due to the acknowledgement section.

Weaknesses of the paper

* The main issue with the paper is that the main text is not self-contained. The proposed method or the rationale of the method cannot be understood by reading the main text. Authors refer to the appendix for most of the important details of the method and the experimental study. Hence, the paper lacks in clarity.

* Further, this paper does not highlight the research gap they try to fill w.r.t to the existing work. At the very end, the authors have mentioned that their method is better than ROAR as their method doesn’t require additional training of the model, but no quantitative comparison is presented.

* They have claimed the following, but it is not supported.
“We demonstrate that, while perturbation artifacts indeed exist, we can minimize and characterize their impact on fidelity estimation by utilizing model accuracy curves from perturbing input features according to the Most Import First (MIF) and Least Import First (LIF) orders.”
In the main text of the paper, it is not clear how you minimize or characterize the impact of perturbation artifacts on the fidelity of the attribution methods.

* The experimental section is also not clearly presented.

* The paper also lacks in reproducibility as the proposed method is not described well. While how to quantify the MIF and LIF graphs is clear, how to derive the fidelity of the interpretability method and impact of perturbation artifact are not clear.


**Suggested Changes:**

*  I would like to suggest authors to stick to one interpretability method and one perturbation method to clearly convey the message without introducing a lot of factors to the picture given that the page limit is just two pages.

* While the authors have tried to solve an important issue in interpretability domain, due to the issues in writing (as mentioned above) their contribution has not been clearly communicated. I believe that if authors rewrite the paper to clearly convey the key message, this paper would have a good impact.

---

### Meta-Review · Area_Chair_Pet1 · 2023-04-06

**Recommendation:** Invite to present (notable)
**Confidence:** 5

**Metareview:**

According to the comments of the reviewers and my own reading, I believe this work is useful for assessing the fidelity of existing attribution-based post-hoc interpretability methods. The experiments are extensive and analysis is solid. The organization and presentation  are fine. A weakness of this work is that the authors are expected to make more analysis in A.1.4.

**Summary:**

This work provides a new method to assess the estimator that determines which features are important.

**Comments And Feedback To The Authors:**

See the meta review.

**Reason For Not Giving A Higher Recommendation:**

N/A

**Reason For Not Giving A Lower Recommendation:**

Novel and solid work. Also, it is clear.

---

> ### Author Response · Authors · 2023-05-30
> **Changes according to reviews**
>
> We thank all reviewers for their helpful comments and recommendations. We have improved our paper by outlining the method for estimating the influence of perturbation artifacts in more detail in the main text. The main text is now fully self-contained, with all variables defined therein. The section A.1.4 has been improved by explaining the motivation and method unverlying the considered important estimators in more detail.

---

### Decision · Program_Chairs · 2023-04-10

Invite to archive